# Rio 2016 Olympic Legacy for Residents of Favelas: Revisiting the Case of Vila Autódromo Five Years Later

## Claudio M. Rocha

Faculty of Health Sciences and Sport, University of Stirling, Stirling FK9 4LA, UK; claudio.rocha@stir.ac.uk

**Abstract:** The aim of this research was to explore and describe the long-term social impact the Rio 2016 Olympic gentrification had in Vila Autódromo from the perspective of former and current residents. Vila Autódromo is a small favela located next to the Rio 2016 Olympic Park. It was almost totally removed during the process of preparing the area to host the Games. In this research, I interviewed 13 residents who passed through the process of eviction threats and displacement. Five still live in Vila Autódromo, whilst eight moved to social apartments provided by the city hall. Interviews revealed that the legacy of Rio 2016 for Vila Autódromo residents can be understood from three broad themes: (1) disempowerment of the community, (2) resistance and resilience during the process, and (3) life after the Games. The residents see the city hall as the main culprit of their displacement, as they were denied their right to the city. However, they also mention the catalytic role of the Olympic Games during the process. They conclude that the legacy of Rio 2016 for them is a very sad story.

**Keywords:** gentrification; human rights; social impact; sport mega-events; sustainable development





## 1. Introduction

Legacy has become an omnipresent discourse in contemporary Olympic affairs (MacAloon 2008). Candidate cities have used the discourse of positive legacy to legitimatise bids to host the Olympic Games. Bid winners maintain such discourse during the preparation-to-host process (Hiller 2014). The International Olympic Committee (IOC) shapes the discourse of legacy on the idea of universal positivity, where legacies are best represented by the benefits left for the host cities (MacAloon 2008; Talbot 2021). The mediatic power of the Olympic "brand" legitimatises that discourse, reinforcing that the Games represent a unique opportunity for a long-term positive social legacy, inclusive to all economic strata in in host communities (Gaffney 2019; Minnaert 2012). The legacy discourse has become quite similar among bidders and hosts. The discourse tries to highlight the long-term benefits for local communities, whilst minimising the negative impacts of hosting. When local communities are negatively affected by event preparation, organisers and event owners are fast to point out that those are short-lived impacts, and that positive legacy takes time to materialise.

The focus of the current research is on the long-term social legacy of Rio 2016 Olympic and Paralympic Games (hereinafter, Rio 2016) for favela residents. I waited until five years after Rio 2016 to revisit the case of Vila Autódromo, a small favela located next to the Rio 2016 Olympic Park. From 2009, when Rio was chosen to host the 2016 Games, until the eve of the event, the city hall tried to remove Vila Autódromo altogether to create space for new constructions in the gentrified area of the Olympic Park (Sánchez et al. 2016). There are reports about what happened with Vila Autódromo before and immediately after the Games (Faulhaber and Azevedo 2015; Ivester 2017; Talbot and Carter 2018). However, when the Games passed, the interest in Rio faded away, moving quickly to the next hosts. Considering the concerns with long-term legacy and sustainable development of host communities (Leopkey and Parent 2012), revisiting the case is important to advance the knowledge about social impacts of hosting the Olympic Games.

Some research has described how forced evictions have happened and have focused on resistance and resilience of residents of Vila Autódromo (Sánchez et al. 2016; Talbot and Carter 2018; Williamson 2017). Experiences of those residents are still fundamental to inform other communities facing the same type of eviction threats. Nevertheless, in the current research, I go one step further and add the perspectives of those who left Vila Autódromo after facing eviction and other types of threats. Some studies have shown that, during the Olympic gentrification, authorities use false information and subversion of facts to threat and persuade residents to agree to leave their houses (Donaghy 2015; Wang et al. 2015). During the preparation for Rio 2016, research has shown that the then-mayor of Rio, Eduardo Paes, used similar strategies. For example, based on the argument that Vila Autódromo was aesthetically and environmentally damaging the area, Paes threatened to remove residents without any compensation (Freire 2013; de Oliveira et al. 2018). Later, the city hall said that the type of ground where the community was settled did not allow infrastructure development[1], thus proposing the relocation of residents to social housing (de Oliveira et al. 2018). Fearing evictions with no compensation, between 2013 and 2014, three-hundred and forty-one families agreed to move to a social housing complex, where the mayor promised excellent infrastructure, leisure, and safety[2] for their families (RioOnWatch 2016; Williamson 2017).

In 2015, with Rio 2016 fast approaching and half of the families refusing to negotiate their leave, the mayor issued a decree that legalised the demolition of many houses under the argument of "public interest" (Sánchez et al. 2016). By March 2016, most of the residents had been either forcibly removed or displaced by fear to social houses. Out of about 700 families, only 20 resisted the multiple attempts of eviction and stayed in Vila Autódromo after the Olympic Games (Williamson 2017). In the current research, I explore the perspective of both groups, those residents who resisted and stayed and those residents who left Vila Autódromo after the actions and pressures from the city hall. The aim of this research was to explore and describe the long-term social impact Rio 2016 Olympic gentrification had in Vila Autódromo from the perspective of former and current residents.

### 1.1. Theoretical Framework: Right to the City

I draw upon the 'right to the city' framework (Lefebvre 1967) to analyse experiences of residents of Vila Autódromo during the process of preparation to host the Rio 2016 and afterwards. Right to the city ("droit à la ville") is characterized by the right of citizens to enjoy urban life and to participate in decisions regarding urban spaces (Lefebvre and Nicholson-Smith 1991). The right to the city involves two principal rights of urban residents: the right to participation (residents play a central role in decisions that contribute to the production of urban space) and the right to appropriation (residents have the right to access, occupy, and use urban space) (Marcuse 2009; Masuda and Bookman 2018; Purcell 2002). The current research produces new knowledge related to the right to participation by directly hearing from residents about their participation in the process of urban generation triggered by the Olympic Games in their neighbourhood. The research also explores the right to appropriation as it investigates how such residents have negotiated the right to stay in their community when gentrification and political forces tried to displace them from their area.

Drawing upon the right to the city, some research in the context of the Olympic Games has proposed that urban regeneration has superficially integrated and directly oppressed marginalised residents of the host cities (Horne 2018; Kennelly and Watt 2011; Watt 2013). I argue that residents of favelas in Rio were not integrated at all, rather they faced constant threats of eviction. To support my argument, I refer to conceptual articles that have proposed that the Olympic-led urban regeneration reinforces the principle of neoliberalism and denies the right to the city to marginalised groups (Hall 2006; Maiello and Pasquinelli 2015). Few empirical articles have applied the concept of the right to the city to marginalised residents (Ivester 2017; Kennelly and Watt 2011). In the current study,

I explored the case of urban regeneration in the specific site where the Rio 2016 Olympic Park was constructed and how this has affected the right to the city of long-term residents.

*1.2. Context: Olympic Gentrification in Rio*

State-led gentrification is at the core of Olympic development projects, which have had impressively similar social, spatial, and financial expressions in host cities as diverse as Beijing, London, Vancouver, and Rio de Janeiro (Broudehoux 2007; Gaffney 2019; Kennelly and Watt 2011; Sánchez and Broudehoux 2013). Neoliberalism has been the institutional rule guiding the gentrification process in Olympic cities (Hall 2006; Rocha and Xiao 2022). Governments have used hosting opportunities to regenerate deprived areas of cities, using neoliberalism to favour capital development, which has led to displacement of poor residents from such areas (Hall 2006; Rocha and Cao 2022; Watt 2013). Displacement of residents to create space for real estate constructions has happened in association with the Olympic Games at least since Seoul 1988 (Rocha and Xiao 2022). Rio 2016 was the most recent example of how governments use the Olympic Games to justify large-scale eviction to create space for new constructions and richer residents (Williamson 2017).

At the beginning of the preparation to host process (2009–2010), the municipal housing secretariat of the Rio city hall listed 119 favelas that must be totally removed by the end of 2012 (Bastos and Schmidt 2010). Favelas are informal settlements that have become the main form of affordable housing in large Brazilian cities such as Rio (Williamson 2017). In Brazil, Rio is the city with the highest number of people living in favelas, with about 1.39 million people (Salles 2021). During the seven-year preparation time (2009–2016), Rio city hall displaced more than 77,000 people from their houses in favelas or poor communities, arguing for the necessity of making space for infrastructural projects that were somehow related to Rio 2016 (Faulhaber and Azevedo 2015; Zimbalist 2017).

*1.3. Vila Autódromo*

Among those favelas threatened to be fully removed, Vila Autódromo became a special case because of its location (less than 300 meters from the entrance to the Olympic Park) and the resistance of its residents. The location may explain the interest of the city hall to remove the community. There was an estimation that new properties in the area of the Olympic Park would cost 200% more after the Games (Cerqueira and Pessoa 2017). Therefore, the area of Vila Autódromo became of great interest for private real estate companies. However, despite multiple attempts to the community, some of their residents were not willing to leave their community and resisted eviction. Some facts about Vila Autódromo explain such resistance.

Vila Autódromo was a fishermen settlement established in 1967, close to the Jacarepaguá lagoon in Rio de Janeiro. Since then, the settlement grew from a few families to about 700 families. Despite the growth, the community created a history of strong social bonds and became a peaceful place (Williamson 2017). Vila Autódromo was not an illegal settlement and residents were not "invaders", as part of the media and the city hall used to refer to them. Most residents have bought their houses and had titles of possession, issued in 1998, by the Rio de Janeiro State Department of Land Affairs; the titles gave them the legal right to occupy the land for a period of 99 years, renewable for more 99 (Williamson 2017). Additionally, in 2005 Vila Autódromo was declared an Area of Special Social Interest by the Municipal Law 74, which established the area as a site for affordable housing construction (Williamson 2017). Some areas of Vila Autódromo did suffer from infrastructural problems, such as lack of integrated sewage, precarious electricity system and lack of road paving. Some authors have asserted that the main cause of the infrastructural problems was the abandonment from the city government, which historically refused to respond to the basic needs of the community (Faulhaber and Azevedo 2015). Despite such infrastructural problems, most of the residents loved to live there because of the sense of community they developed over the years (Faulhaber and Azevedo 2015; Williamson 2017).

As part of Rio 2016 hosting preparation, attempts to remove Vila Autódromo started in 2009, when the proposal was to construct the Olympic media centre in the area. When the media centre was moved to another site, the city hall said that the community still needed to be removed because of the security perimeter of the Olympic Park (Coaffee 2015; Williamson 2017). The security perimeter argument was not sustained because a large number of new apartment buildings were constructed in the area as part of the gentrification plan. Additionally, the British architecture company AECOM, the bid winner responsible for designing the Olympic Park, proposed a project to urbanise the community as part of the Rio 2016 legacy (Comitê Popular do Rio 2013). Despite that, the city hall made multiple attempts to remove Vila Autódromo during the process of preparation to host.

The city hall had a neoliberal agenda, which created benefits for private real estate companies. Some of those companies were against having favelas in the new gentrified areas, affirming that that would be detrimental for their businesses. There are media reports where real estate owners expressed their views on how poor communities did not "fit" well in the new gentrified area of the Olympic Park (for example, see Puff 2015). To explain what was going on in Rio, some authors proposed that the city government had created a "state of exception" (Agamben 2005), where laws were adapted and human rights were ignored to promote the interest of private capital (Aaron Richmond and Garmany 2016; de Oliveira et al. 2018). Vainer (2011) asserted that, during the preparation for Rio 2016, Rio was a "city of exception", where everything was permitted to support a neoliberal agenda.

*1.4. From Favelas to Social Housing*

During later stages of the preparation to host the Olympic Games (2013–2016), the case of Vila Autódromo increasingly gained international notoriety. Stories of human rights abuse started to appear in United Nations reports and more frequently in international media (Rolnik 2015; Wilkson 2016). Therefore, to manage international pressure and avoid political loss, Rio authorities changed their strategies from accusing residents of being invaders of the area to trying to persuade them to agree to leave their houses, which would be exchanged by money compensation or apartments in social housing complexes (Donaghy 2015; de Oliveira et al. 2018). Based on this strategy, the construction of a close-by social apartment complex named Parque Carioca, to where residents could opt to move in, was the single most important fact to convince residents to leave Vila Autódromo.

Parque Carioca was part of the federal government housing programme "Minha Casa, Minha Vida" (my house, my life), which provided affordable opportunities for low-income people to buy their first property. Parque Carioca was the only known case where houses/apartments from "Minha Casa, Minha Vida" were offered to people who already had houses, breaking a fundamental rule of the programme (de Oliveira et al. 2018). During the preparation for Rio 2016, the political parties of the Rio city mayor (Eduardo Paes) and the president of Brazil (Dilma Rousseff) were allies, facilitating the use of a federal programme to support the mayor's agenda. The political use of that federal programme was clear and it has been criticised by other authors (de Oliveira et al. 2018; Williamson 2017).

Parque Carioca was strategically located to tackle the criticism that displaced residents are moved to houses very far away from their communities and jobs (Faulhaber and Azevedo 2015; Sánchez and Broudehoux 2013). It was offered to residents with many attractive features, such as a swimming pool and playground areas for children (Robertson 2016). The argument of the government was that they were offering a much better way of living for favela residents, in a very close-by area, which would not disrupt their lives. Considering the context—a vulnerable community, tired of years of threats of eviction and lack of infrastructure—it is not difficult to understand why many residents accepted the city offer and moved to the Parque Carioca. In 2015, there was an estimation that 75% of the Vila Autódromo residents had agreed to move out, either by taking a money compensation or by accepting another place to live; among those, 32% of the residents had moved to Parque Carioca (Faulhaber and Azevedo 2015; de Oliveira et al. 2018). In this research, I

aimed to hear from those former residents who agreed to move to Parque Carioca, in order to understand their thoughts and perspectives about what happened during the process and their current situation.

Despite the government pressures and incentives, by 2015, about 25% of the families were still not willing to leave the community (de Oliveira et al. 2018). Therefore, a year before the Olympic Games, the Rio government escalated their actions to persuade residents to agree to move. Such actions included daily visits from government officials saying that everybody would have to leave and the last ones would leave with no compensation at all (de Oliveira et al. 2018). The demolition of the houses of those residents who agreed to move was another strategy of the government. Lastly the government started promoting forced evictions of those who had not yet agreed to leave (Williamson 2017). In that context, most of the residents that had resisted up to that point agreed to take the government's compensation and left the community (Faulhaber and Azevedo 2015). By March 2016, with the Olympic Games around the corner, the city hall had no more time to implement additional tactics to remove them; then, the Rio mayor agreed that the 20 families that were still in Vila Autódromo could stay and the city hall would provide the urbanisation of the area to settle them (Williamson 2017). It was a victory for those families, representing a victory for the community. In this research, I also aimed to hear from those who stayed, in order to understand the process that they passed through and what created such resilience and determination to stay.

## 2. Method

I collected data in June/July 2021, five years after Rio 2016, using semi-structured interviews. After getting ethics approval, I started contacting some potential interviewees. I have been part of the Sustainable Favela Network, which is an initiative of the Catalytic Communities, a local non-governmental organisation whose mission is to generate opportunities for community-led development in Rio favelas. After finding the first residents/former residents interested in participating, I used a snow-ball sampling strategy, where some participants indicated the next ones. At the time of data collection, Brazil was encouraging social distance due to the COVID-19 pandemic. Therefore, rather than face-to-face, I conducted the interviews online. To define the number of interviews, I applied the principle of theoretical saturation (Guest et al. 2006). The interviews were conducted with 13 people, eight former residents and five residents of Vila Autódromo. The inclusion criteria were that the resident must be at least 26 years old (or 21 years old in 2016, during the Games) and have lived in Vila Autódromo for at least eight years (enough time to create social ties). Table 1 shows some demographic information about interviewees and the time they had lived in Vila Autódromo when the interviews were conducted. The interview scripts had questions that allowed participants to express their thoughts and perspectives about what happened with them during the time of preparation to host Rio 2016 and what long-term social legacy they have perceived the event has left to them.

To analyse the data, interviews were voice recorded and fully transcribed in Portuguese; then, quotations that were used in the article were translated to English. I used NVivo 12 to undertake an iterative coding exercise and identify key themes, following six steps of thematic analysis (Braun and Clarke 2006): (1) immersion in the data through intensive reading; (2) generation of initial codes; (3) search for themes; (4) revision of themes; (5) definition and naming of themes, and (6) report writing.

**Table 1.** Characteristics of participants.

| Participant | Gender | Residence | Years in Vila Autódromo |
|---|---|---|---|
| Resident 1_PC | Female | Parque Carioca | 20 |
| Resident 2_PC | Female | Parque Carioca | 19 |
| Resident 3_PC | Female | Parque Carioca | 21 |
| Resident 4_PC | Male | Parque Carioca | 22 |
| Resident 5_PC | Male | Parque Carioca | 15 |
| Resident 6_PC | Female | Parque Carioca | 16 |
| Resident 7_PC | Female | Parque Carioca | 26 |
| Resident 8_PC | Female | Parque Carioca | 26 |
| Resident 1_VA | Female | Vila Autódromo | 21 |
| Resident 2_VA | Male | Vila Autódromo | 21 |
| Resident 3_VA | Female | Vila Autódromo | 21 |
| Resident 4_VA | Female | Vila Autódromo | 30 |
| Resident 5_VA | Female | Vila Autódromo | 20 |

## 3. Results and Discussion

The aim of this research was to explore and describe the long-term social impact Rio 2016 Olympic gentrification had in Vila Autódromo from the perspective of former and current residents. Interviews revealed that the legacy of Rio 2016 for Vila Autódromo residents can be understood from three broad themes: (1) disempowerment of the community, (2) resistance and resilience during the process, and (3) life after the Games. Residents also discussed the role of Olympic Games and their owners, the IOC, during the process.

### 3.1. Disempowerment of the Community

The fact the Olympic Park had an agenda and was about to be built in the area close to Vila Autódromo created an urgency from Rio city hall. To fulfil the agenda, the city hall applied strategies that can, according to reports of residents, be divided in three groups: psychological pressures (based on arguments of public interest and intense gentrification in the area—to create feelings of not belonging), money compensation (through offers of "better" places for residents to live and/or cash in exchange of their houses), and violence (using the law and establishing a frequent police presence in the favela, carrying out forced evictions). The aim of the strategies was to disempower the community to carry out the total removal of Vila Autódromo.

Residents told me that the psychological pressures were quite intense during the preparation-to-host period. At the beginning of that period, residents were told that they must leave to create space for the Olympic venues. A former resident reported that, "They told us that we must leave because they would build the Olympic Park in the area; then, we must leave, no option" (Resident 7_PC). At that point, they suffered eviction threats based on the argument of "public interest". A former resident exemplified the type of pressure they were exposed to:

We suffered a very intense psychological pressure. When everything started, we got a letter from the city hall saying that the place where our house was located was a place of public interest. Then … the psychological pressure started. They put down some houses around us, that shook our structure (Resident 6_PC).

Public interest is a broad, diffuse, and fuzzy concept that can be used by governments to impose their interests over individuals' interests (Lewis 2006). The concept has been one of the main arguments to evict low-income residents from areas where the government has economic interest in different parts of the world, such as Africa (Ocheje 2007), Latin America (Santoro 2019), and South Asia (Bhan 2009). Ocheje (2007) defends that the public interest rationale is flawed and proposes that evictions of marginalised residents are better explained by inappropriate urban planning and government corruption. In Rio, the area of Vila Autódromo became an area of "public interest" for the city hall because of the Olympic Park and all real estate speculation that was generated around it. The government used the argument of public interest to impose a psychological pressure under residents, who were



told that they were blocking progress and preventing the city to create a "greater good" for more people. Apparently working in the best interest of all, the government was actually defending the capital interest and trying to create a new zone to be explored by private real estate companies (Gaffney 2016; Sánchez et al. 2016).

The gentrification itself was a form of psychological pressure to lead residents out of the area. It sent a message to the residents that they did not belong to that region anymore. A resident from Vila Autódromo nicely summarises the idea of gentrification as a strategy to remove:

When a place is abandoned, the poor can stay. They can go there and create a favela, build some shacks . . . the city hall does not care at all. However, when the place starts to improve, when the city hall starts to improve the infrastructure of the place, then you cannot stay there anymore. You feel like you do not belong there. But you have arrived there much earlier . . . (Resident 5_VA).

The same feeling has been reported from residents in other Olympic cities. For instance, Watt (2013) reported how gentrification of the London borough of Newham (to host London 2012) created a feeling of not-belonging among long-term residents. This idea is linked to Marcuse's (1985) indirect displacement or displacement pressure, which occurs when people see their neighbourhood changing dramatically—friends leaving the region, new ways of access, local shops closing, new unfamiliar shops opening. Such changes create a feeling of not belonging where you had once belonged. The ultimate impact of such feeling is to push long-term, marginalised residents away, to create space for new, richer residents that are supposed to occupy the gentrified area (Marcuse 1985).

Psychological pressures were accompanied by money offers for residents to leave their houses. Money compensation happened in two forms: The city hall offered an apartment in Parque Carioca in exchange for their houses or cash in their bank accounts for them to buy a new place where they preferred. When the Games were far away, the city hall offered an exchange between their houses in Vila Autódromo and an apartment in Parque Carioca. A resident told me that, "They wanted the vast majority to go to the apartments that they had built as part of the 'Minha Casa, Minha Vida' programme" (Resident 5_VA). At this stage, the city hall used different strategies to convince residents to accept the exchange. For instance, "They used to take families to visit the apartments by van. They furnished the apartments with very expensive furniture [ . . . ] showed the swimming pool, sporting courts for kids [ . . . ] it was the fantasy land for favela residents" (Resident 5_VA). Considering the lack of structure those families had in Vila Autódromo, it is not difficult to understand why 32% residents (or about 200 families) agreed to move to Parque Carioca (Faulhaber and Azevedo 2015; de Oliveira et al. 2018). Although this migration was very positive for the government plans, that was not so positive for those residents who moved out. Below I present the view of those who accepted to move into Parque Carioca.

At this stage, the government also offered cash in exchange for their houses, because some people did not want to move to apartments. Some people took the cash and moved to other areas of the city. Despite different strategies adopted by the city hall, many families still refused to leave their homes in Vila Autódromo. Then, with the Olympic Games fast approaching, the city hall adopted a new strategy. As a resident remembered, "From 3 June 2015, the city hall started a new process. They started offering both an apartment *and* cash compensation" (Resident 1_VA, emphasis in her speech). Another one said, "But when they started offering apartment and cash compensation, many people decided to go. Each one with their reasons. Whether we agree or not, we have to respect and understand" (Resident 8_PC). Using this strategy, the city hall managed to remove many more families. At the same time, large parts of the community were, at this point, in ruins because the city hall was fast to destroy the houses of those who agreed to leave. However, as Williamson (2017) noted, not everyone in Vila Autódromo had a price. About 25% of the residents were still not willing to leave their community (de Oliveira et al. 2018).

From psychological pressures and money compensation the government escalated the strategies and started to adopt a more violent approach to force residents to leave.

This transition was perceived by a resident who said, "Then, the harassment was getting worse. They started demolishing the houses of those who left. They brought the city police. That was terrifying for us because the community was a peaceful and calm community" (Resident 5_VA). To legitimise the use of violence to remove, the government started the process of land expropriation. Still in 2015, supported by a federal decree-law from 1941[3], the city hall carried out land expropriation of an area where 58 houses were located in Vila Autódromo, claiming to that the area was a zone of "public interest". Some people who were part of the process claim that it was not by chance that the houses of some of the leaders of the resistance movement were included among the ones to be removed (Williamson 2017). This illustrates the final stage of disempowerment of the community. To remove families that had refused to leave, the city hall used the decree-law to implement forced evictions.

The use of force pushed more residents out of Vila Autódromo. A former resident told me how it made her decide to move out: "There was no need of so many police in a such small community. It was a form of intimidation. I felt highly intimidated. I was extremely afraid. The fear made me to move out" (Resident 2_PC). Another person who also moved out reported her experience:

I was not physically assaulted, but my son was. He got a lot of rubber bullets. [ . . . ] That was on the day when they tried to evict an elderly man from his house in the community [ . . . ]. He got injured. A friend of mine [ . . . ] suffered a very serious injury when a police officer broke her nose [ . . . ] It was an extremely difficult moment. Everything was very impactful and aggressive (Resident 2_PC).

*3.2. Resistance and Resilience during the Process*

Despite psychological pressures, money compensation and violence, some residents kept resisting the idea of moving out of Vila Autódromo. Residents described some factors and agents that motivated them to resist and develop resilience during the battle to stay in their community. Some mentioned that long-term feelings of social injustice have given them motivation to resist the displacement attempts during the preparation for Rio 2016. A resident said, "What made me fight to stay was my conscience, the need to fight against that type of social injustice" (Resident 4_VA). Another one added that, "I started realising the injustice [ . . . ] we have lived here for more than 20 years, I grew up here, I raised my family here [ . . . ] why do I need to leave my home for an event that lasts 17 days?" (Resident 5_VA).

Those feelings of injustice seem to bind them tightly and increase the sense of community. A former resident, who resisted for a long time before leaving, said that her strength to fight came from social bonds created within the community: "My strength to fight came from living there [ . . . ] when you have people that you can count on at the worst moments, you found a treasure" (Resident 8_PC). Another one said, "We were like a big family [ . . . ] The friendship was excellent over there" (Resident 4_PC). This sense of community was also found in other communities who resisted evictions and displacements. For instance, Watt (2013) described a similar sense of community among residents of Carpenters Estate, in the borough of Newham, during the preparation of the site to host the London 2012 Olympic Games. Although the cultures of Brazil and the UK are very different, the similarities between Vila Autódromo and the Carpenters, in the way that community ties had fostered resilience, are remarkable.

Being aware of the strong positive ties of the community, according to some residents, the city hall used some strategies to destroy such ties. A resident said, "They [the city government] bought some residents, whose job was to try to convince others to go to Parque Carioca" (Resident 2_VA). Another one said, "A very cruel thing that happened during that time was that they split the community. [ . . . ] They recruited some people to convince others that everybody would have to leave. [ . . . ] That was very sad because they were our neighbours [ . . . ]" (Resident 5_VA).

Despite the existence of those internal problems, residents also mentioned some internal and external agents who helped them to be resilient during the process. Initially, they remembered the support from internal agents. Vila Autódromo has a catholic church—Parish "São José Operário"—that was mentioned by many interviewees as a great point of support during the fight to stay. A former resident said, "The catholic church offered much support. People who had their houses demolished after the decree stayed at the church, living there. That was a major support we had within the community" (Resident 6_PC). The man who was leader of the church for part of the process, Father Fabio Guimarães, was also mentioned by many residents as a fundamental agent for community resilience. For instance, a former resident informed that:

Father Fabio was on our side at all the times. He used to say for us not to leave, because it was our right to stay [ . . . ] He was very intelligent. He explained everything to us, in a very clear manner. Many times, people came and said things that we did not understand, trying to shuffle our ideas. But then Father Fabio came and explained the things to us (Resident 2_PC).

It was impressive how many residents spontaneously mentioned the figure of Father Fabio as a point of support within the community. He seemed to have played a very important role during the resistance to stay. Therefore, a fact that undermined the resilience of the community was the transfer of Father Fabio to Rome, Italy. A former resident remembered, "When they took him, it was a shock for me and for the whole community. Wow, how come Father Fabio will abandon us! Everything was very fast, very dark. I felt abandoned" (Resident 2_PC). Another one said, "We thought very strange that he was transferred to Rome. We still do not know the reasons, but he was with us in the fight [ . . . ] We stayed like orphans. Everything happened very fast" (Resident 3_PC). Some residents speculated that Father Fabio might have been transferred because of his community activism, as he had been encouraging residents to stay. The activism created additional hindrances for the plans of the city hall of removing the community altogether. In interviews, residents agreed that his transfer was a heavy blow to the community, affecting their resilience.

When discussing resilience, residents also mentioned support from external agents. A resident summarised this by saying that "We had support from many people, from public attorneys to universities, researchers [ . . . ] to international media. Then, we felt empowered and said to ourselves, let's keep going" (Resident 4_VA). The role of local universities was mentioned by some residents. They remembered with gratitude what a group of faculty members from two public universities in Rio did to increase their resilience: "We had the 'Popular Plan for Urbanisation of Vila Autódromo', developed by Fluminense Federal University and Federal University of Rio de Janeiro in collaboration with local residents" (Resident 1_VA). The Popular Plan for Urbanisation of Vila Autódromo was a scientific report where both universities detailed how Vila Autódromo could be upgraded (including sewage, lighting, road paving, leisure areas, and affordable houses) under a modest budget (Vainer and Oliveira 2018; Williamson 2017). Residents felt supported by the universities and empowered by the plan because it gave them a document that was saying that the eviction of the community was not necessary. The quality of the plan was recognised by external organisations (e.g., the London School of Economics) and it won an international award sponsored by the Deutsche Bank (Ivester 2017). Despite that, the city hall refused to put the plan in action.

Public attorneys also received multiple mentions when residents talked about the support they received. A resident said, "Some public attorneys defended our legal rights. [ . . . ] Their commitment was extremely important for us. [ . . . ] We had highly committed attorneys by our side" (Resident 5_VA). Another one complemented this, saying that, "They [the public attorneys] kept saying that the law was on our side and that, at any moment, if someone tried to remove us, we should call them immediately" (Resident 2_PC). The support of external agents to help communities to develop resilience to face tragedies and disasters is not unique to Vila Autódromo. However, it is worth noting that the literature

usually places the government as one of the main external agents that support community resilience (Imon Chowdhooree 2020; Platts-Fowler and Robinson 2016). In the case of Vila Autódromo, the government was on the other side. Therefore, other external agents assumed the role of external supporters. Beyond local universities and public attorneys, residents also mentioned other external agents, such as non-governmental organisations (NGOs) and the media. Some NGOs were nominated in the interviews (e.g., Catalytic Communities and "A Pública") and were acknowledged as great source of support for resilience development during the process.

Regarding the media, during the preparation for Rio 2016, the mainstream media had a narrative that described favelas as a "hindrance" for the city to materialise the full potential of the Olympic "legacy" (Bastos and Schmidt 2010). Therefore, it was somehow surprising when residents described the media as a source of support for their resilience. However, residents distinguished between the mainstream media and alternative media. A resident said, "The mainstream media used to destroy us, they used call us 'of invaders' [ . . . ] but the alternative media, neighbourhood newspapers, internet sites, small channels, they supported us. Through them we had a counter narrative against the big media" (Resident 5_VA). When talking about the mainstream (big) media, residents were referring to open TV channels and popular newspapers, which have regularly reported favelas through the lenses of stigmatised stereotypes of poverty, drugs, and violence—and a stumbling block for the "success" of the Olympics. A better understanding about favelas show that that is a big mistake based on clear prejudice (Romero 2015; Santiago 2017). For example, data inform us that only a small percentage (about 2%) of favela residents are involved with drug dealers; whilst it is true that some drug cartels have been able to flourish in some favelas, that has been caused mainly by the abandonment that the state impose to favelas (Romero 2015). Scholars argue that favelas are not the problem, rather, for decades, they have been the solution to provide affordable houses for low income workers (Williamson 2017).

Beyond the role of the mainstream media, residents cited other facts that show how the city hall tried to undermine the resilience of the community. A resident said, "The mayor went to extreme measures, such as suspending the delivery of mail and cutting trash collection. We were living in the middle of debris. We also used to stay without electricity and water [ . . . ]" (Resident 4_VA). This created a sequence of shocks in the community, fitting well the concept of "shock therapy" as described by Klein (2007). Shock therapy is a sequence of brutal tactics with the aim of disorienting people and ultimately advancing neoliberal agendas. Examples of "shocks" have included wars, terrorist attacks, market crashes, and natural disasters (Klein 2007). In Vila Autódromo, shocks included the use of not only direct and illegal tactics, such as cutting electricity and trash collection, but also indirect (and sometimes not so obvious) ones, such as disempowerment of leaders, splitting the community, and the use of mainstream media to damage their image. Creating shocks was the tool to destabilise the order of the community and create a dire need of change (Klein 2007). That was the Rio city hall tactic in Vila Autódromo. Klein calls this "disaster capitalism", where shock therapy is used to promote neo-liberal agendas. This is not new in the context of preparation for hosting the Olympic Games (Boykoff 2014; Lenskyj 2008; Watt 2013), although I would argue that, in the context of Rio 2016, the shock therapy reached another stage when compared to previous Games. The facts described by Vila Autódromo's residents provide support for my argument.

Hayes and Horne (2011) suggested that sport mega-events are "the apparently benign twin of disaster capitalism's shock therapy" (p. 752). Because of the celebratory nature of sport events, Boykoff (2014) suggested that instead of disaster capitalism, the Olympic Games have pushed forward the "celebration capitalism", which he identifies as "the affable cousin" of disaster capitalism (p. 3). Boykoff proposes that celebration capitalism succeeds on social euphoria, not on shock therapy. The Olympic Games represent the most successful case of celebration capitalism because of social euphoria and festival atmosphere that hosting the Games is able to create (Boykoff 2014). Results of the current study show that, in the case of Vila Autódromo, there are elements of both disaster (through shock

therapy) and celebration capitalism (through social euphoria). Eviction and destruction of houses created a "war" atmosphere in the community, indicating a shock therapy. "Then, you see only emptiness and a lot of debris. It was very hard. We felt like in a war zone. I do not want to compare, but ... We were terrified" (Resident 8_PC). Meanwhile, in a few meters of distance, the main stage for the spectacle of the 2016 Olympic Games was being constructed, feeding the social euphoria about hosting the most visible sport event in the world for the first time in the Global South. In this context, the fight to stay in Vila Autódromo became a unique case of resistance against both disaster and celebration capitalism.

*3.3. Life after the Games*

The literature has provided evidences for the relationship between hosting the Olympic Games and displacement of marginalised residents (for a review, see Rocha and Xiao 2022). However, studies in the literature are limited to report impacts in short periods before or after the event (e.g., Kennelly and Watt 2011; Shin and Li 2013). I have not found studies reporting long-term impacts. This is an important gap in the literature about social legacies. In this section, I present how those who left and those who stayed in Vila Autódromo describe life a long period after the Games.

Most of the former residents of Vila Autódromo reported some regret for having moved out. Complaints appeared multiple times in different interviews. They are related to three major issues: loss of community interaction, broken promises, and financial burden. Some described how they have lost and missed community interaction. A resident said:

I miss it [Vila Autódromo] a lot. Sometimes I even cry. Now I live alone in an apartment. You know, when you live in an apartment, you do not have the same interaction with your neighbours [ ... ] you cannot sit at the door, offer a coffee, chat [ ... ] In apartments, everyone enters, closes the door [ ... ] you rarely see other people (Resident 2_PC).

Although Vila Autódromo and Parque Carioca are relatively close to each, the lifestyle changed dramatically for those who moved to Parque Carioca. Previously, the literature has criticised the fact that displaced people are usually sent to locations that are far away from their original places, creating problems to have access to their jobs and/or schools and to keep social ties (Greene 2003; Zheng and Kahn 2013). In the case of Vila Autódromo, the city hall sent people to a close-by social apartment complex. Whilst this may not have created problems for residents to have access to their jobs and schools, this has created problems for them to keep their social ties.

There was also some disappointment with broken promises. For instance, a former resident says that "they said that they would build a creche and a family clinic here [in Parque Carioca], but nothing was actually made" (Resident 4_PC). Most of the residents told me that many of the promises of the city hall were not kept. Some mentioned promises related to help with the maintenance of the apartments. Others said that there were even promises of helping with condominium fees. The fact is that residents have struggled moving from a place where they paid their own utility bills to a condominium-like apartment complex, where on top of their own bills they still have other taxes to pay.

The financial burden gets worse when the city hall created a mortgage in the name of people who moved into Parque Carioca. This appears in multiple interviews as a major issue of regret and resentment. A resident explains what happened:

Some people have mortgage arrears here in Parque Carioca. I am one of those. This is frustrating because we did not buy this apartment, we exchanged it by our house. That was the deal. However, they [the city hall] created a mortgage in our name. Now, when they do not pay, our names go down to debtor lists[4] (Resident 1_PC).

Another resident confirmed the problem: "The mortgage is in arrears. [ ... ] The name of my wife is in the debtor lists" (Resident 4_PC). Residents have raised this issue with the city hall, which says that the mortgage has been paid. Then, they tried to argue with other institutions, but this has only created more frustration because those institutions seem not to have instruments to help. For instance, a resident said: "Suddenly they stop paying.

Then, I receive a letter from the bank. I went there and said, *I am not responsible for paying this, the city hall is*. Then, the bank says, *but the mortgage is under your name*". (Resident 5_PC). Some residents told me the city hall eventually pays the mortgage in arrears. However, the delay in the payment has already created troubles for credit scores of people who have not requested any mortgage.

Other residents still raised the problems created by the fact that the apartment has not been fully paid by the city. A resident summarises this quite well:

It was not an exchange, there is a mortgage. [ . . . ] I did not enter in any programme to get a new house. I received a letter saying that I must leave my house [ . . . ] the mayor said that it would be an exchange, that we would be able to sell or do whatever we wanted with our apartment. [ . . . ] The reality is that we cannot do anything because we have a mortgage. [ . . . ] I try to argue with the city hall [ . . . ] but they always sound like they have done a big favour for us, taking us out of a community that had no pavement, no sewage, to place us here in this apartment complex (Resident 6_PC).

In the sport mega-event literature, I have not found reports about what happened to residents who were displaced from their houses in the long run. However, the feelings of frustration and resentment that former residents of Vila Autódromo reported seem to be similar to those found in internally displaced persons by wars, disasters, and economic crises (Crisp et al. 2012; Kett 2005). In the case of these extreme events, people move out of their houses after uncontrollable events. Despite the fact these events are usually unpredictable, this still creates feelings of frustration and resentment. In the current study, there was no uncontrollable or unpredictable extreme event forcing residents out of their houses. Hosting of sport mega-events is planned years ahead, so are the urban transformations necessary to host. Participants in this study were quite aware of this. Therefore, it is not surprising that they demonstrate an increased feeling of frustration and resentment, five years after moving, when the broken promises and the financial burden still persist.

There are also losses for those who resisted and stayed in Vila Autódromo. A resident provided an excellent summary of these losses:

The poor is the one that loses the most. In that event, only 20 families stayed in the community. Only 20 families [pause to breathe]. In my point of view, it was a big victory for those 20 families, we won our permanence. We stayed in the same territory. But the community and the place changed a lot. They cut many, many trees. We used to have many streets, today we have only two streets. All this is very hard. We lost almost the whole community—from 700 to 20 families (Resident 2_VA).

Residents who stayed do not hide the pain of having lost the close contact with family members, friends, and neighbours. All of them expressed tones of nostalgia in their comments. They all talk about the good moments in the past, when the whole community was still there. Despite the losses, a former resident noted that those who stayed and even some who moved out share a feeling of victory: "I am happy that the community survived. Even though only 20 houses are part of the community now, the community resisted and survived. We won the battle. I was also part of this fight" (Resident 6_PC).

*3.4. IOC Turns a Blind Eye*

The residents recognised the Rio city government as the main culprit by the displacement they suffered. However, an interesting finding of the current study is related to how some interviewees understand the role of the Olympic Games in that process. Many interviewees reported that the proximity of the event created an urgency for evictions and displacements. For instance, a resident said, "the evictions effectively started during the period [of preparation] for the Olympic Games. The Olympics Games made it possible" (Resident 4_VA). Other residents acknowledged the value of the Games, but they still see that the benefits are not for all. "The Games are a cool thing. It brings visibility to the country. However, unfortunately, they are not used to the benefit of all. They benefit some and harm many" (Resident 3_PC).

A resident associated the problem of evictions and displacement not only with the event, but also with owners and organisers. She said:

In the case of the Olympic Games [ . . . ] the IOC and the organisers, they knew about the situation. They are not innocent. They were conniving. [ . . . ] They are as oppressive as the city hall, maybe even more. They pretend to be blind. [ . . . ] Are you going to tell me that they did not know that there was a community near to the Olympic Park with more than 700 families?" (Resident 3_VA)

There are reports saying that IOC was informed about the situation. For example, Williamson (2017) informs that the Land and Housing Department[5] of the state of Rio de Janeiro sent an eighty-page document to the IOC, describing human rights violations that were happening in Vila Autódromo. Williamson says that, at the end of 2010, the IOC asked the Rio de Janeiro State then-governor, Sérgio Cabral, about the situation. The governor replied saying that problem was resolved. To resolve the problem, the governor shut down the department and sent the public attorneys who prepared the document to other posts (Williamson 2017). The IOC has not followed up on the situation.

Years later a journalist wrote to the IOC asking about the evictions and displacements that had happened in Vila Autódromo during the preparation to host Rio 2020 (Donahue 2020). He received back a written statement that says, "The displacement was not dictated by the needs related to the hosting of the Olympic Games. This was a decision of the city [of Rio] itself and the IOC made it clear at the time that these displacements were not needed for the Games to take place" (Donahue 2020, p. 22).

The IOC seems to have a missed a unique opportunity to show that Olympic Games can deliver a real social legacy for local communities. They could have supported the maintenance and urbanisation of Vila Autodromo and shown to the world that they actually care about human rights and are against evictions and displacement of marginalised citizens. Unfortunately, this has not been the legacy for Vila Autódromo. As a resident summarises, "The legacy that the Olympic Games left for us, in Vila Autódromo, was a very bad legacy. It was a legacy of disgrace, of stories of lives that were destroyed [ . . . ] [For us] The history of the Olympic Games is very sad" (Resident 2_PC).

## 4. Conclusions

Results show that the social legacy of Rio 2016 for Vila Autódromo's residents was negative for both groups—those who stayed and those who left. Residents who stayed reported that they lost their community and close contact with friends and family. The geography of the area was also totally modified, green areas were destroyed, and new houses for those who stayed were built. Although they stayed in the territory, the community and the space were certainly not the same. Residents who left reported a feeling of deep regret for having accepted to exchange their homes for apartments, which have created financial and social distress for them. They live in a place with a better infrastructure when compared to their previous houses in the favela. However, they show regret and resentment for having accepted the exchange. They report isolation, frustration, and resentment due to the loss of community interaction. Whilst they point to the city hall as the main culprit of the destruction of the community, they also report that the Olympic Games created a momentum to disempower them to the point of being evicted or displaced. Therefore, they consider their situation as a legacy of the Rio 2016 Olympic Games.

Results of the current research confirm conceptual articles that have proposed that the Olympic-led urban regeneration reinforces the principles of neoliberalism and denies the right to the city to marginalised groups in the city (Hall 2006; Maiello and Pasquinelli 2015). Residents of Vila Autódromo reported that they have not had the right to participate in decisions about the urban space where they lived, nor have they had the right to keeping occupying a space that was already theirs. They had scientific and professional support from universities and public attorneys to claim that their permanence in the area was legal, safe, economically viable, and sustainable. Despite such support, the city hall denied their right to the city.

**Funding:** This research has received no external funding.

**Institutional Review Board Statement:** The study was conducted in accordance with the Declaration of Helsinki, and approved by the Institutional Review Board (or Ethics Committee) of University of Stirling—GUEP (19 20) 935 on 27 July 2020.

**Informed Consent Statement:** Informed consent was obtained from all subjects involved in the study.

**Data Availability Statement:** Data is available upon request, respecting the data Protection legislation in the UK, governed by two main pieces of legislation, the UK General Data Protection Regulation (UK GDPR) and the Data Protection Act 2018 (DPA 2018).

**Conflicts of Interest:** The author declares no conflict of interest.

## Notes

[1] The mayor's argument was contested by experts in urbanism from two local universities—"Universidade Federal do Rio de Janeiro" and "Universidade Federal Fluminense", which developed an urban plan showing how infrastructure improvement of Vila Autódromo would be possible in that type of ground, without affecting the environment. This plan had direct contribution of residents. For more information about this plan, see (de Oliveira et al. 2018).

[2] Safety was not an issue in Vila Autódromo, as participants of the current study confirmed in their interviews. Despite the stereotypes that associate favelas with lack of safety, other authors have also affirmed that Vila Autódromo was a safe place, with no drug cartels or armed militias (Faulhaber and Azevedo 2015; Talbot and Carter 2018; Williamson 2017).

[3] The federal Decree-Law 3365 (from 21 June 1941) allowed any level of the Brazilian government to expropriate any land or house for public interest. Based on this decree, the mayor emitted a municipal decree (39,853 from 18 March 2015) expropriating some of the houses in Vila Autódromo.

[4] In Brazil, there are two organisations that provide credit scores. These are "Serviço de Proteção ao Crédito" (SPC) and Serasa Experian. These are the names residents mentioned when talking about going to lists of debtors.

[5] Land and Housing Department is a free translation for "Núcleo de Terras e Habitação" (NUTH).

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
