# Peer review of "Rio 2016 Olympic Legacy for Residents of Favelas: Revisiting the Case of Vila Autódromo Five Years Later"

_socsci, doi:10.3390/socsci12030166_

Round 1

Reviewer 1 Report

The authors of this paper study the impact of the 2016 Olympic Games in Rio, specifically examining the impact of gentrification in the neighborhood of Vila Autódromo. The paper is generally very well conceived and executed, especially in terms of the research question  and the concrete nature of its methodology and research findings. The authors interviewed former and current residents of the neighborhood, those who stayed and those who left. They find that he impact is negative and lasting. Such findings indeed show the dark side of sports in very clear and specific ways. The authors relate their findings appropriately to gentrification and the perspective of right to the city. A very good contribution. (typo in abstract: "The residents see city hall has the main culprit", should be "as the main culprit").

Author Response

Thank you for your word of support for my research. Thank you also for pointing to the typo in the abstract. It is now corrected. 

Reviewer 2 Report

In the introduction, the authors explain very well the context. Their intention is to study the impacts of the heritage of the Olympic Games in RIO and more precisely in a favela close to the games. The authors question the consequences of the gentrification of the spaces bordering the Olympic Games. The evaluation of the social impact of this global event is a very interesting subject and deserves our attention. 

The theoretical framework (right to the city) questions the right of residents to be able to participate in local life and local political decisions. The authors adopt an activist posture protesting against neo-liberalism and the expulsion of marginalised populations. This posture, far from the axiological neutrality of the scientist, is however assumed. They want to denounce the process of using the Olympic Games to remove and expel poor people for the benefit of the more privileged. Far from solving the problem, this only moves it elsewhere. The authors propose to interview the populations concerned. The authors refer to a great many authors and works. This is a good thing. The authors describe well the situations of psychological pressure and violence to expel the occupants. The text is instructive and the reader can become aware of the deplorable effects of the JO on the populations, especially the poorest. This reality deserves to be shown and published.

However, one main criticism appears. As part of the population has been settled in new districts, and others have refused to leave, the authors propose to interview the populations concerned (8 displaced people and 5 stayed). The methodology is clearly presented, but the choice to interview only people from the favelas implies a "biased" reading of the subject studied, reinforcing and confirming the authors' very militant posture. The authors would also be well advised to interview public actors to find out their opinion, because the principle of objectivity is strongly questioned.

I think this text is very interesting and could be published if the authors agree to balance their point of view and adopt a less militant posture.

Author Response

Thank you for your review. We really appreciate it. Please, see attached a pdf file with our answers. 
